# Natural Intra- and Interclade Human Hybrid Schistosomes in Africa with Considerations on Prevention through Vaccination

**DOI:** 10.3390/microorganisms9071465

**Published:** 2021-07-08

**Authors:** Ursula Panzner, Jerome Boissier

**Affiliations:** 1Division of Infectious Diseases and Tropical Medicine, Ludwig Maximilian University of Munich, 80539 Munich, Germany; 2Swiss Tropical and Public Health Institute, University of Basel, 4002 Basel, Switzerland; 3IHPE, University of Montpellier, CNRS, Ifremer, University of Perpignan, 66860 Perpignan, France; boissier@univ-perp.fr

**Keywords:** hybridization, introgression, *Schistosoma*, immunization, Africa

## Abstract

Causal agents of schistosomiasis are dioecious, digenean schistosomes affecting mankind in 76 countries. Preventive measures are manifold but need to be complemented by vaccination for long-term protection; vaccine candidates in advanced pre-clinical/clinical stages include Sm14, Sm-TSP-2/Sm-TSP-2Al^®^, Smp80/SchistoShield^®^, and Sh28GST/Bilhvax^®^. Natural and anthropogenic changes impact on breaking species isolation barriers favoring introgressive hybridization, i.e., allelic exchange among gene pools of sympatric, interbreeding species leading to instant large genetic diversity. Phylogenetic distance matters, thus the less species differ phylogenetically the more likely they hybridize. PubMed and Embase databases were searched for publications limited to hybridale confirmation by mitochondrial cytochrome c oxidase (COX) and/or nuclear ribosomal internal transcribed spacer (ITS). Human schistosomal hybrids are predominantly reported from West Africa with clustering in the Senegal River Basin, and scattering to Europe, Central and Eastern Africa. Noteworthy is the dominance of *Schistosoma haematobium* interbreeding with human and veterinary species leading due to hybrid vigor to extinction and homogenization as seen for *S. guineensis* in Cameroon and *S. haematobium* in Niger, respectively. Heterosis seems to advantage *S. haematobium*/*S. bovis* interbreeds with dominant *S. haematobium*-ITS/*S. bovis*-COX1 profile to spread from West to East Africa and reoccur in France. *S. haematobium*/*S. mansoni* interactions seen among Senegalese and Côte d’Ivoirian children are unexpected due to their high phylogenetic distance. Detecting pure *S. bovis* and *S. bovis*/*S. curassoni* crosses capable of infecting humans observed in Corsica and Côte d’Ivoire, and Niger, respectively, is worrisome. Taken together, species hybridization urges control and preventive measures targeting human and veterinary sectors in line with the One-Health concept to be complemented by vaccination protecting against transmission, infection, and disease recurrence. Functional and structural diversity of naturally occurring human schistosomal hybrids may impact current vaccine candidates requiring further research including natural history studies in endemic areas targeted for clinical trials.

## 1. Schistosomiasis and Parasite Hybridization

### 1.1. Disease and Transmission

Schistosomiasis affects mankind in 76 countries causing an estimated annual mortality rate of 280,000, 779 million people at risk of infection, 250 million people with active infections, and 440 million people with residual morbidity [1,2,3,4]. The disease burden is largest throughout sub-Saharan Africa accounting for >90% caused by the clinically most relevant human-pathogenic species *Schistosoma mansoni* (Sm) and *S. haematobium* (Sh) [1,2,3,4]. Though schistosomiasis is likely underrecognized among the veterinary sector, inclusive of livestock and wildlife animals, it is assumed that in particular ruminants, rodents and non-human primates are carriers of zoonotic species [5]. 

The causal agents of this neglected tropical disease are dioecious digenean schistosomes within the Platyhelminthes; species by clade of the *Schistosoma* genus including their geographical distribution, and intermediate and definitive hosts are illustrated in Figure 1 [5,6,7,8] with supporting data in Appendix A. Figure 1 illustrates that the spectrum of snail intermediate hosts is restricted to a limited number of species of a given *Schistosoma* genus, whereas the spectrum of definitive hosts varies from a single host, e.g., *S. intercalatum* (Si) infecting humans only, to multiple hosts, e.g., *S. japonicum* (Sj) infecting humans and animals. 

The parasite species-specific snail-to-vertebrate transmission is mediated by skin-penetrating cercariae during contact with infested freshwater that transform into schistosomulae and migrate matured to adult worms to their oviposition sites within the vascular system for mating. While adult worms of most parasite species live in the blood vessels surrounding the intestine, soley Sh adults persist in the perivesical vascular system. Schistosomes are monogamous, thus one female is fitted per male gynaecophoric canal, though mate changes are reported allowing homo- and/or hetero-specific inter- and/or intra-species worm crossing in the hepatic portal system [9,10]; ova produced are released via feces or urine to continue the vertebrate-to-snail transmission for asexual reproduction upon miracidia hatching into freshwater, or retained in tissues causing parasite species-dependent inflammatory pathologies due to granulomatous-fibrotic formations [3,11,12,13].

**Figure 1 microorganisms-09-01465-f001:**
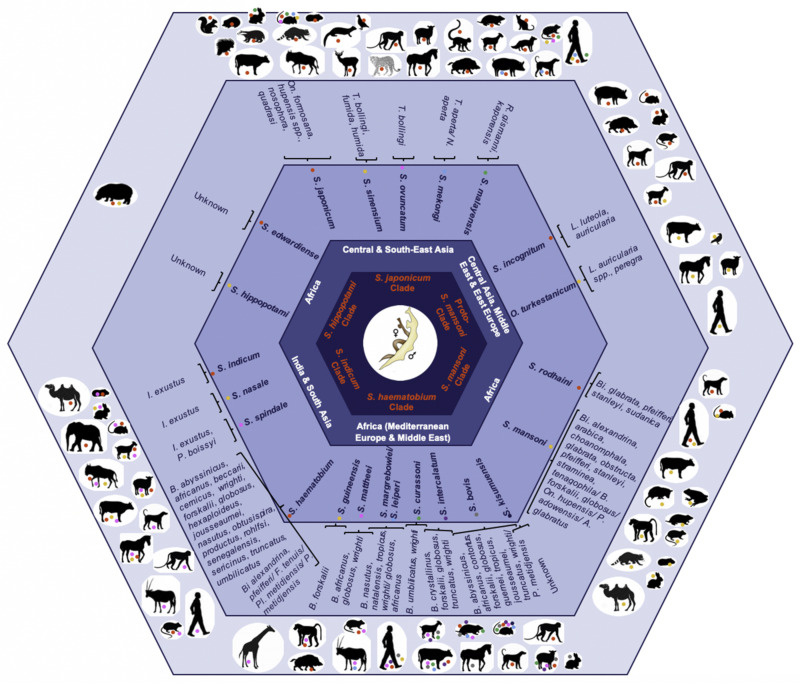
Clades within the *Schistosoma* genus inclusive of their geographical distribution, *Schistosoma* species, and species-specific intermediate invertebrate and definitive experimental and natural mammalian hosts ordered from the inside to the outside of the hexagon. Notes: *S.* = *Schistosoma, O.* = *Orientobilharzia*; *On.* = *Oncomelania*, *T.* = *Tricula*, *N.* = *Neotricula*, *R.* = *Robertsiella*, *I = Indoplanorbis*, *P.* = *Planorbis*, *L.* = *Lymnaea*, *Bi.* = *Biomphalaria*, *B.* = *Bulinus*, *A.* = *Australorbis*, *Pl.* = *Planorbarius*, *F.* = *Ferrissia*; adapted from Leger et al. [5], Lawton et al. [6], Webster et al. [7], and Pearce et al. [11]; species-specific intermediate and definitive hosts retrieved from the host-parasite database of the Natural History Museum [8].

### 1.2. Treatment and Prevention

The chemotherapeutic at present is the acylated quinoline-pyrazine known as praziquantel (PZQ) that acts against adult schistosomes though poorly against immature schistosome larva by changing irreversibly the permeability and stability of tegumental membranes and needs established host immune defense mechanisms for full efficacy [2,3,14]. Anti-worm IgA, IgE and IgG1, IgG2 and IgG3 subclass immunoglobulins are expressed subsequent to treatment resulting in protection against re-infection lasting 6–12 months, and IgG4 subclass immunoglobulins leading to higher susceptibility to re-infection due to IgE blocking [15,16]. However, IgG4 antibody levels decrease with regular repeated treatment. Serious rebound morbidity and emerging resistance are pressing concerns due to PZQ’s exclusive use since its discovery [17].

Preventive measures are multifaceted, but need to be complemented by vaccination to achieve long-term protection against transmission, infection, and disease recurrence as acquired naturally in endemic settings [1,18]. The goal is a non-sterilizing immunity with long-term decline in tissue eggs and egg excretion preferably through killing of female worms while preserving concomitant natural immunity induced by less-pathogenic single male worms [19,20]. Since schistosomes do not replicate in definitive hosts, consensus exists that a prophylactic product should reduce adult worm burden among vaccinated, and egg excretion among infected individuals each by at least 75% [4,16]. To date, no vaccine is commercially available also due to known immunoevasion and modulation mechanisms of adult schistosomes, and limited understanding of impacts of recurrent exposures, poly-parasitism, co-infections, and chemotherapeutic treatment on the host immune system [16].

Although, few promising candidates have advanced in their preclinical and clinical stages, e.g., Sm14 in Phase 2a/2b, i.e., Sm fatty acid-binding protein (FABP), Sm-TSP-2/Sm-TSP-2Al^®^ in Phase 1b/2b, i.e., Sm tetraspanin, Smp80/SchistoShield^®^ in progression of entering Phase 1, i.e., Sm large-subunit calpain, and Sh28GST/Bilhvax^®^ in Phase 3, i.e., Sh glutathione S-transferase [19,21,22].

FABPs are required for the uptake, transportation and compartmentalization of host lipids due to lack of schistosomal oxygen-dependent pathways to synthesize long chain fatty acids and cholesterols de novo [23]. Large homologies of FABPs’ amino acid sequences with greatest residue conservation among the C-terminal molecule are reported for e.g., *S.j.*, *Echinococcus granulosus*, *Clonorchis sinensis*, and *Fasciola hepatica*, a liver fluke of global importance among humans and livestock causing tremendous economic losses [24,25,26,27]. This demonstrates the candidate’s potential for cross-species multipurpose protection which is of relevance for human and veterinary applications.

Tetraspanins are scaffold proteins involved in cell biology and physiology, trafficking and functioning of membranous proteins, and immunoregulatory as well as immunoevasion processes through adsorption of host molecules for masking the parasite’s “non-self” status [28,29]. All tetraspanins share a conserved topology of four hydrophobic stabilizing transmembrane domain proteins (TM1–4), a short highly conserved intracellular N- and C-terminal loop, and two protein–protein interacting extracellular mushroom-like loops, i.e., the small EC1 domain of 17–22 amino acids, and the large EC2 domain of 70–90 amino acids with conserved steam and variable head regions; phylogenetic polymorphism among EC2 alters the affinity and avidity to hosts’ immune responses, and stimulates varying levels of protective efficacy similar to var-genes of *Plasmodium falciparum* [30,31,32,33].

Calpain as an amphitropic, heterodimeric proteolytic protein consists of the 30 kDa regulatory subunit that activates the 80 kDa catalytic subunit through a cascade of calcium-activated auto-proteolyses; four calcium-binding EF hand motifs among domain IV of schistosomal calpain constitutes the calcium-binding region [34]. Though the number of amino acids constituting calpain seems the same among *Schistosoma* species, functional and structural differences through amino acid substitutions exist as seen for Sm and Sj [35]. Calpain is expressed in all schistosomal lifecycle stages though with 2.5-fold higher intensity on the female tegumental lipoidal membranous outer-surface bilayer, and plays a pivotal role in tegumental biosynthesis and turnover, a helminthic mechanism to evade host immune responses [36].

Glutathione S-transferase is central in regulating detoxification, antioxidant pathways and fatty acid metabolism, immune modulation during infection, e.g., migration of Langerhans and dendritic cells, and neutralization of host hydroperoxides [37]. Its crystal structure consists of two similar monomers, each having N- (residues 4–87) and C-terminal (residues 88–207) domains; 75% of all residues including N- and C-terminal domains are conserved among e.g., Sm, Sj and *S. bovis* (Sb) while 28 kDa GSTs of Sh and Sb exceed residue conservation due to their close evolutionary relationship leading to GST’s capability of a cross-species vaccine candidate [38].

### 1.3. Instant Gene Exchange

Hybridization is defined as the uni- and/or bi-directional allelic exchange among gene pools of sympatric interbreeding species infecting the same hosts, including repeated backcrossing with parental species; it leads to large genetic diversity of novel inter-species and/or inter-lineages trough instant acquisition of genetic material in contrast to immigration or mutation [5,39,40,41,42]. Parasite hybridization is an emerging public health concern and may be linked to global natural and anthropogenic changes causing selection pressure; this in turn impacts the dynamics and distribution of schistosomes and their hosts by breaking isolation barriers, thus increases chances of introgressive hybridization [2,40,43,44,45].

As human migration increases, people encounter new infections more frequently, and co-infections with different parasite species within the same individuals augment. Natural hybridization of human-pathogenic agents has been shown in protozoan, fungal, viral, bacterial [41], and helminth [42] systems. Compared to free-living animals, parasites are often hidden and have limited morphological characters or other identifiable phenotypes, thus the frequency, and ecological, evolutionary importance of parasite hybridization is largely underestimated. However, schistosomes are known to hybridize and natural hybrids are more and more frequently found disorienting baseline laboratory and clinical diagnoses; see Figure 2 for likely natural schistosomal hybridization events across Africa [46]. The probability of introgressive hybridization with production of viable progeny or parthenogenesis depends on the phylogenetic distance of Schistosoma species, thus species of low distance within the same monophyletic clade are more likely to exchange their gene pools than species of high distance [5,44,47]. Consequently, ShxSm hybrids seen in Senegal and Côte d’Ivoire were unexpected [40,47,48].

Recombinants are characterized by heterotic alterations, speciations, neo-functionalization and adaptations, called hybrid vigor [49]. This affects e.g., virulence, transmission, infectivity, pathologies, maturation, fecundity, host spectrum and preference, e.g., extended host range of Shx*S. mattheei* (Sma) hybrids to sheep, and ShxSb hybrids to non-human primates and rodents, and chemotherapeutic efficacy, e.g., diminished response to oxamniquine if Sma interacts with Sh [5,13,39,44,50,51,52,53,54,55]. Taken together, this may result in the expansion to new geographic habitats, and homogenization or progressive, competitive extinction of species as seen for *S. guineensis* (Sg) in Cameroon, and ultimately the emergence of new diseases [49].

The occurrence of ShxSb crossbreeds in Europe with a likely West African origin was decisive for an in-depth investigation of published reports on natural human hybridale schistosomes among databases of PubMed and Embase in addition to considerations on prevention of schistosomiasis through vaccination based on aforementioned leading candidates. Hybridization of schistosomes could be suspected by ectopic egg elimination, e.g., excretion of Sm eggs also in the urine, or visualization of the egg morphology, i.e., eggs having an intermediate morphology between two species. However, both are not sophisticated markers. Ectopic eggs could be indicative of alternate routes of excess egg excretion, and as the egg shape of first generation F1 hybrids and mother species is identical hybrid ancestry cannot be confirmed either [56]. Ultimately, only molecular characterization can assess the hybrid status of parasites with certainty as seen based on the first natural schistosomal hybrids detected in the region of Lake Victoria, Tanzania [57]. Therefore, articles included in this investigation were limited to hybrid confirmation by mitochondrial cytochrome c oxidase (COX) and/or nuclear ribosomal internal transcribed spacer (ITS), both commonly used in phylogeny and phylogeography, and due to assumed poor sensitivity of other confirmatory tests; see Figure 3 for more details [58,59,60]. ITS, as a genus-specific marker, distinguishes hybrids and their backcrossed progeny as parental sequences retain for generations before getting homogenized by concerted evolution; COX is a haploid-inherited mitochondrial marker for species-specific identification evolving more rapidly than nuclear genes [2,39,44,46,56,61]. Hybrids are detected either when the biparentally inherited ITS marker shows heterozygous pattern at specific mutation points and/or if ITS and COX markers give discordant results in species assignation. Recent developments in single nucleotide polymorphisms (SNPs) markers on both, Sh and Sb, will certainly advance the characterization of genomic properties of hybrids in the near future [49].

## 2. Schistosome Hybrids across Africa—The Tip of the Iceberg

### 2.1. S. haematobium/S. guineensis Hybrids

Interactions between Sh and Sg are reported from South-West Cameroon; of note is the re-description of *S. intercalatum* (Si) from Lower Guinea, e.g., Cameroon, Gabon, Equatorial Guinea and São Tomé, as Sg [50,51,62,63,64]. Ecological changes such as forest clearance and agricultural development evolving in the 1960s besides population movement favored the presence of snail hosts, e.g., *Bulinus wrighti, B. forskalii* and *B. truncatus*, and species sympatry and interplay; this materialized the complete replacement of Sg by Sh within 25–30 years through introgressive hybridization and competitive extinction abetted by behavioral, reproductive and genetic advantages [65,66,67]. Experiments confirmed greater competitiveness and pairing abilities of ShxSg crosses over parental species, and Sh being dominant over Sg, which supports observations made in Cameroon and elsewhere, e.g., Kinshasa, Democratic Republic of Congo, and Dogon Country, Mali [68,69,70,71]. Retrospective experiments among schistosomes of laboratory maintained urinary ova of Cameroonian children detected hybridization between Sh and Sg among 100%, 33% and 5% isolates from Kumba, Loum (1990), and Loum (1999 and 2000), respectively; the remainder had pure Sh, but no Sg despite established *B. forskalii* hosts [62]. Findings were confirmed by four-banded single-stranded conformational polymorphism (SSCP) profiles of second nuclear ribosomal internal transcribed spacer (ITS2) fragments. Sequencing isolates from Loum revealed unequivocal intermediate SNPs at the positions 25, 80 and 130 within ITS2 fragments compared to parental strains, i.e., guanine-cytosine-guanine for Sh, and adenine-thymine-adenine for Sg, clearly demonstrating ShxSg recombinants with dominance, thus inheritance, of Sh over Sg [51].

Schistosomes of laboratory maintained urinary and fecal ova of residents in Melen and Ekouk, West Gabon, both settings with established *B. forskalii* and *B. globosus* hosts, were examined to assess presumed interbreeding events of Sh with species causing intestinal schistosomiasis along the road axis of Libreville-Lambaréné [72,73,74]. Despite SSCP’s capability of detecting single point mutations, all ITS2 profiles demonstrated pure Sh schemes compared to parental strains from Benin/Niger and Cameroon, respectively. No recombinants were found though ova, possibly deposited ectopically due to high infection rates, were isolated from urine and stool samples [39,75]. Of note is that research from Gabon and Cameroon has laboratory passages of miracidial hatching from field ova with maturation to adult schistosomes in common, which may limit their meaningfulness as this procedure is prone to artificial selection of specific parasite genotypes [44,76].

Urinary and fecal specimens from schoolchildren in Doh and Dangbo, Benin were studied due to Sg expanding its geographical distribution from Lower Guinea to West Africa, including Nigeria, Mali and Burkina Faso; while Sh is widely spread across the Beninese population, Sg seems absent [50,77]. High-resolution DNA melting analyses performed on miracidia hatched solely from urinary eggs revealed homozygote ITS2 patterns for 26% and 72% Sh, and 23% and 5% Sg, but heterozygote patterns for 51% and 23% ShxSg in Doh and Dangbo, respectively; sequencing confirmed double chromatogram peaks and SNPs guanine/adenine, cytosine/thymine, guanine/thymine at the polymorphic positions 25, 80 and 130. Partial mitochondrial cytochrome c oxidase subunit 1 (COX1) fragments showed 6.6% base pair substitutions compared to parental isolates, and 0.7–0.9% and 5.7–5.9% base pair substitutions when aligned to Nigerian Sh and Cameroonian Sg male worms, respectively. This indicates introgression of Sg genes into the Sh genome with Sh as the maternal lineage. Male Sg must have crossed with female Sh if effective competition among males of both species, and active choice of the oviposition site by females can be assumed.

### 2.2. S. haematobium/S. bovis Hybrids

ShxSb hybrids are certainly the most widespread and best studied schistosome hybrids in West African countries. The presence of these hybrids was first discovered in 2009 in the Senegal River Basin (SRB) [44] and attributed to contemporary ecological and anthropogenic changes having affected the SRB since the 1980s. It was pointed out the construction of the Diama Dam on the Senegal River for sea water desalination and expanded irrigation and agriculture, and the Manantali Dam on the Bafing River for hydroelectricity and alterations in water velocity; *Biomphalaria pfeifferi* and *Bulinus* spp. hosts of Sm, Sh and Sb got established due to more favorable habitats triggering increases in urogenital and intestinal schistosomiasis in both, humans and livestock, intensified by growing contacts between humans and companion, domestic/livestock and wildlife animals [44,71,78,79,80,81]. These changes have certainly favored the propagation of ShxSb hybrids in the SRB, though recent genome-based studies did show that hybridization events are much older than presumed. Based on parasites recovered from Niger, it has been estimated that introgression occurred around 240.6 years ago (range: −107.8 to −612.5 years) [82]. The ancestral origin of these hybrids was confirmed with parasites originating from several settings across Western Africa [49].

In Senegal, numerous studies have been performed on ShxSb hybrids following their initial discovery. Investigations on miracidia hatched from urinary and fecal ova of Senegalese schoolchildren from six villages in the SRB adjunct to Senegal River, Lampsar River, and Lac de Guier revealed 15% ShxSb isolates with Sh-ITSxSb-COX1 profiles producing viable offspring; patterns were confirmed by double chromatographic peaks at polymorphic positions of ITS and sequence divergence of COX1 with alignment to Sh and Sb strains from Guede Chantier and St. Louis, Senegal, respectively [44,48,56]. Recombinants evolved from crossing of male Sh with female Sb were capable of infecting humans and both *B. globosus* and *B. truncatus* snails indicating loss of host specificity [44]. The reverse, more rare recombinant seems less viable due to biased homogenization towards parental species, asymmetric backcrossing, or concerted evolution. Restriction fragment length polymorphism (RFLP) was applied to COX1 and ITS amplicons of miracidia hatched from urinary and fecal ova of children and adolescents in Pakh, Northern Senegal, and 10 villages of the SRB, followed by back-crossing of fertile hybrid progeny with parental species [48,56]. Approximately 30% Sh-ITSxSb-COX1 hybridale genotypes, 70% pure Sh genotypes, and no pure Sb were reported in 2017. In contrast, only 9% Sh-ITSxSb-COX1 or Sh-COX1xSb-ITS profiles were detected in 2018; interestingly, 0.8% ShxSm-COX1 and 0.4% ShxSm-ITS loci were scattered across sites despite their high phylogenetic distance. Spillover or switching to rodent and other wildlife hosts for zoonotic transmission routes as likely reasons for species interactions observed urges control measures targeting human and veterinary sectors in line with the One-Health concept [10,44,54,83,84]. Further interactions between Sh and Sb also with *S. curassoni* (Sc) were seen among intestinal worms, and hepatic and urinary miracidia from domestic livestock and schoolchildren, respectively, from Northern SRB, central Valé du Ferlo, Southern Tambacounda, and South-Eastern Kolda, Senegal [45]. Of the human miracidia, 79% were pure Sh, whereas 21% had ITS/COX profiles indicative of ShxSb and ShxSc first generation hybrids due to crossing of male Sh with female Sb or Sc; host switching facilitating species interactions, or initial pairing in rodents for instance producing viable offspring likely occurred. Of the livestock worms, neither pure Sh nor interbreeding with either species were seen, except for few SbxSc recombinants among slaughtered cattle; however, no urinary specimens were collected limiting the meaningfulness of findings though Sh’s inability to penetrate and develop in animals is assumed anyhow. Among miracidia hatched from urinary ova of adult hospital patients from Richard-Toll, lower SRB, 63% had pure Sh, and 37% Sh-ITSxSb-COX1 or Sh-COX1xSb-ITS profiles identified by double chromatograms at the polymorphic sites [80,85]. The area, characterized by high population fluctuations due to seasonal workers of food-processing sectors from adjunct countries, was exclusively prevalent for intestinal schistosomiasis, i.e., Sm, until the early 2000s. Hybrid cases detected are of particular interest as adults are commonly exempted from preventive mass drug administration programs, and represent a potential likely unnoticed source of ongoing transmission.

Though Southern Europe was regarded as parasite-free since the 1950s–1960s, ShxSb crossbreeds were found among urinary ova of a 12-year-old boy from Corsica, France; he likely got infected during recreational activities near Corsican rivers, e.g., Cavu River, during a large schistosomiasis outbreak during summer 2013 [75,86,87,88,89]. Isolates revealed three haplotypes among COX1 amplicons with 0.21% sequence divergence, and 98.43–99.48%, 93.19–94.97%, 88.90–90.16% and 98.22–98.85% sequence identities with Sb, Sc and Sh, respectively, from Sô-Tchanhoué and Toho, Benin, and Melen and Ekouk, Gabon, and GenBank data. Three haplotypes detected among ITS2 fragments showed 99.98% sequence identity; haplotype-1 and -2 harbored Sh SNPs, i.e., guanine-cytosine-guanine, at the polymorphic positions 25, 80 and 130, whereas haplotype-3 harbored a single SNP, i.e., adenine, of either Sb, Sc or Sg at position 130. Pairing of male Sh with female Sb and subsequent backcrossing with parental Sh likely explains the majority of COX1 and ITS2 patterns observed [49]. Of note is the shared Sh-ITS2xSb-COX1 behavior between Sô-Tchanhoué and Corsica strains though it does not explain its recurrence despite established *B. contortus*, *B. truncatus* and *Planorbarius metidjensis* hosts throughout Southern Europe [90,91]. The initial introduction of ova through an infected person from an endemic region during summer 2013, when environmental conditions were favorable, is assumed [91,92,93] as the parasite incriminated showed ShxSb hybrid pattern of Senegalese origin [86]. This genetic evidence can be corroborated by sociocultural information as Senegal and France share a close relationship for touristic and economic purposes. Young Northern Senegalese visit France since decades for occupational opportunities in the gastronomic sector, e.g., the restaurant Sainte-Lucie de Porto Vecchio about 15 km from the bathing sites of the Cavu River, which does not constitute formal proof, but helps to better understand the likely parasitic spread. However, data available do not explain parasitic survival during less favorable winter months, and recurrence in subsequent years beyond 2014; a potential zoonotic role of livestock and/or rodents, the latter being capable of serving as hosts for Sb and ShxSb crossbreeds, but not Sh, as seen in the SRB and recent Beninese investigations is suspected [54,81,90,94,95,96,97]. Interestingly, Savassi and colleagues did confirm the presence of ShxSb hybrids in cattle and rodents (*Mastomys natalensis*) in addition to schoolchildren in Kessounou, Benin [97,98] in contrast to research performed in Senegal [45] and Cameroon [99]. The possibility of zoonotic parasite transmission to human hosts would derange dramatically well-established prevention and control measures for schistosomiasis. However, investigations of Oleaga et al. did not detect parasitic evidence among surveyed cows, goats or sheep. Rodent trapping in the vicinity of Cavu River revealed a single rat infected with a single worm harboring the Sb2 haplotype, which is frequently found in Corsica and also Senegal [93]. Although access to Cavu River was denied in 2014 as an immediate action of epidemic control, recurrent outbreaks were detected during summer months throughout 2015–2018 [100]. In addition, further parasite spread from Cavu River to Solenzara River, situated 20 km in the North, was observed [100]. We also found investigations on urinary ova or miracidia from subsequent Corsican cases that confirmed Sh-ITS2xSb-COX1 hybridale profiles, but also pure Sh and Sb [86]. Detecting pure Sb among humans is unexpected and could have occurred through silencing of parental genes during backcrossing, thus the likelihood of a true zoonotic transmission has to be considered with caution [49]. Minimum spanning haplotype network analyses uncovered six Sh COX1 haplotypes, of which one was identical to the most common dominant haplotype on mainland Africa; likewise, eight Sb COX1 haplotypes were revealed of which one was identical to the haplotype recovered from humans infected with ShxSb recombinants in Senegal, though it clustered away from animal-derived Sb haplotypes. This again suggests a West African origin of schistosomes found in Corsica. Interestingly, the most frequent haplotype, i.e., Sb2, found in Corsica and during recurrent outbreaks on the island is also the most frequent haplotype found in Senegalese settings. In addition to investigations on the role of livestock and/or rodents as potential reservoirs, the capability of infected molluscs to overwinter the temperate island climate was assessed [101]. The study concluded that if molluscs can survive a single winter season, parasite persistence throughout 2013–2018, and/or spread across adjunct rivers cannot be explained, thus reseeding is the more likely explanation.

In Niger, crossing of sympatrically occurring closely related Sb and Sc with COX1/ITS and 18SDNA (partial nuclear ribosomal 18SDNA) patterns, respectively, was found among urinary miracidia from a 10-year-old child from the Tillaberi Region [43,45]. In addition, pure Sh and ShxSc-ITSxSb-COX1 recombinants were detected likely due to repeated introgression and backcrossing of Sh with Sb or Sb interbreeding with Sc followed by mating with Sh [43,45]. The capability of veterinary schistosomes infecting humans directly is of high concern. Of archived miracidia from urinary ova of Nigerien but not Tanzanian children assessed by exome capture and whole-genome sequencing with alignment to reference templates of species within the Sh clade, 65% Sh miracidia had Sb mitochondrial DNA with 3.3–8.2% alleles identified as Sb haplotypes [82]. The length of haplotype blocks demonstrated that introgressive hybridization reached almost fixation within Sh, which likely occurred multiple recombination events ago [102]. Crossing events did not occur contemporarily since an expected 50% or 25% Sb genome among F1 or F2 recombinants was not found [102].

Recent investigations in Côte d’Ivoire identified ShxSb hybrids in both, molluscs and humans [86,103]. Sequences of miracidia collected from schoolchildren at Sh and Sm endemic sites in the south of the country, i.e., Agboville, Adzopé, Sikensi and Duekoué, uncovered 35% pure Sh, and very few pure Sb suggesting zoonotic transmission similar to reports from Corsica [86,103]. Larval stages collected from molluscs at aquatic habitats in the north of the country revealed approximately 52% sequences with interchanged genotypes of Sh-ITSxSb-COX1 pattern, which is the most frequent hybrid type reported with ambivalence in snail choice, i.e., *B. truncatus* and *B. globosus*; seven Sh and 12 Sb haplotypes were detected among the COX1 regions indicative of introgressive hybridization of Sh by Sb.

Ongoing disease surveillance among children residing in Chickhawa, Nsanje and Mangochi Districts of Malawi, lacking reported evidence of Sb, found primarily pure Sh ova and few eggs with discordant Sh-ITSxSb-COX1 schemes [71,104,105,106,107,108]. In addition, ShxSma-18SDNAxSma-COX1 profiles (Sma = *S. mattheei*) were detected similar to presumed ShxSma recombinants seen for instance in Lusaka, Zambia, and Transvaal, South Africa [71,104,105,106,107,108].

### 2.3. S. haematobium/S. mansoni Hybrids

Even before Boon et al. reported miracidial ShxSm-COX1 and ShxSm-ITS patterns, interbreeding between phylogenetic distant Sh and Sm was observed among miracidia collected from schoolchildren in the SRB, Senegal; based on microsatellite DNA amplification, 36% urinary and 2% fecal isolates presented with the hybridale allele profiles 218:239 or 218:236, respectively, whereby allele 218 is specific to Senegalese Sh, Sb and their hybrids, and allele 239 is specific to Senegalese Sm [76]. RFLP applied on COX1 and ITS amplicons of miracidia and ova detected heterozygote Sm-COX1xSmxSh-ITS and Sh-COX1xSmxSh-ITS schemes. Similar to observations made for ShxSb crossbreeds in this setting, a potential zoonotic role of livestock and/or rodents, the latter being capable of serving as host for Sb and ShxSb crossbreeds but not Sh, as seen in the SRB is suspected.

In Côte d’Ivoire, ShxSm hybrids were not detected before, except for travelers returning from the coastal country that sought healthcare in hospitals in France. Viable miracidia of urinary but not fecal ova of a male child with a stay in Divo, Côte d’Ivoire during summer 2007 revealed 25 equally intense, double chromatographic peaks among DNA amplicons using the 28DNA (partial nuclear ribosomal 28SDNA) marker; they were identical to polymorphic positions of Sm and Sh reference templates from GenBank clearly indicating crossbreeding [47]. COX1 showed 99.1% sequence homology with Sm revealing their maternal inheritance. Eggs isolated from urine likely occurred due to hybridization of female Sm with male Sh, which is unexpected due to their high phylogenetic distance. However, as co-infections due to species sympatry with male competitiveness for females, and males determining the oviposition site are reported, natural heterologous interactions are possible [109,110]. Similarly, ova from urinary and fecal samples of a 14-year-old male child from Côte d’Ivoire presented with double chromatographic ITS profiles identical to Sm and Sh, and COX1 haplotypes specific to Sh or Sm that clustered with Sm also from Niger, Senegal and Mali, thus an isolate that likely originated from West Africa [111].

### 2.4. S. mansoni/S. rodhaini Hybrids

Long-term anthropogenic alterations largely affected also the Rift Valley Region in West Kenya leading to SmxSr (*S. rodhaini*) interbreeding due to directional introgression from male Sr into female Sm with ascendance of Sm. Laboratory passaged miracidia hatched from fecal ova and matured schistosomes were compared for their patterns in ITS, 16S-12SDNA (partial nuclear mitochondrial 12S-16SDNA), and additional microsatellite loci to Sr strains from Burundi and Kenya, and Sm from Puerto Rico. Sympatrically occurring species likely interacted because prezygotic reproductive barriers of differential host use, preferential mating, male–male competition with male Sr dominating male Sm, and circadian cercarial emergence from snail points are imperfect [39,112,113].

## 3. Schistosomal Hybridization—Interference on Leading Vaccine Candidates

Extensive small-scale gene amplifications and alterations are contributing to a steady schistosomal genomic evolution [55]. SNPs emerging through transversional/transitional or synonymous/non-synonymous nucleotide substitutions leading to e.g., alternative splicing or silent mutations are of high importance in particular among sequences serving as drug or vaccine targets; they may lead to structural and functional protein alterations impacting charge changes and residue conservation affecting drug binding sites or antibody recognition of antigenic epitopes [27,114,115]. Hosts react in the following general manners against schistosomal infections, i.e., development of age-dependent partial protective immunity to reinfection from repeated adult worm death, and initiation of immunopathogenic and/or immunoregulatory mechanisms against parasitic antigens released from eggs trapped in tissues [116,117,118].

Preliminary findings of the phase 2a trial confirmed rSm14+GLA-SE’s safety and strong, long-lasting immunogenicity, and reported 92% seroconversion following the second dose with cellular responses, memory cells and T-cell activation makers [119]. However, Sm14 isoforms demonstrate the potential of structural instability resulting in its aggregation and precipitation and, in turn, to uncertain immunoprotection and functional alterations [115,120]. SNPs as seen among natural schistosomal hybrids and artificially constructed Sm14 mutants lead to augmented structural integrity without compromising its protective immune response allowing to administer the vaccine candidate in endemic tropical and subtropical settings [55]; depending on the mutant, protective responses ranging from 20% to 67% with mean worm reductions of 20–40% were observed in 4–6 weeks old female outbred Swiss mice. The potential extent and longevity of rSm14+GLA-SE’s immunogenicity against Sm-based hybrids requires future investigations. Of note with this regard is that rSm14+GLA-SE underwent clinical testing in the SRB, particularly the valley area along the Senegal River commencing from Saint-Louis, that seems a wide and stable hybrid zone, i.e., phase 2a dose-escalation, safety and immunogenicity rSm14+GLA-SE trial among healthy male adults with history of Sm and Sh infections receiving single-dose PZQ for parasite clearance with subsequent three-dose immunization; phase 2b safety and immunogenicity trial among schoolchildren both, healthy and infected with Sm and/or Sh, receiving single-dose PZQ pre-treatment with subsequent three-dose immunization. Findings from both trials pending to be published.

rSh28GST+Alhydrogel/Bilhvax^®^ vaccine candidate was tested in a phase 3 safety, efficacy, disease recurrence and immunogenicity trial among Sh infected schoolchildren from the ShxSb and ShxSm hybrid endemic SRB receiving two-dose PZQ pre-treatment, three-dose immunization, and a single-dose PZQ treatment prior to a final vaccine booster dose [121,122,123,124]. rSh28GST+Alhydrogel proved its safety, strong, long-lasting immunogenicity, and anti-worm fecundity with inhibited egg viability despite the fact that the primary endpoint of significant delay in schistosomiasis recurrence was not reached in the trial [121]. Selection pressure may favor the conservation or modification of functional important amino acid residues of schistosomal GST resulting in altered host immune recognition. Early investigations revealed unusual high evolving of its conserved N-terminal domain leading to the assumption that schistosomal 28 kDa GST functions no longer as an enzyme, but rather as a target of the host immune system, in particular for eliciting immune responses impacting the parasites’ fitness [125]. Amino acid motifs of surface proteins repeated multiple times leading to highly polymorphic molecules, called smoke screens, resemble a parasitic strategy of stimulating the host immune system in an ineffective manner [126]. Inter-species differences in evolutionary patterns among the N-terminal domain of the molecule were observed among species of low phylogenetic distance, e.g., Sb and Sh, and high phylogenetic distance, e.g., Sm and Sj, which have diverged about 2-fold in the number of non-synonymous nucleotide substitutions leading to diminished immunogenicity [127]. Polymorphism among immunogenic regions of GST within natural schistosomal parasite populations may impact its capability as a vaccine candidate [127]. Therefore, natural history studies seem of particular importance in areas with known endemicity for hybrid schistosomes targeted for clinical trials. Prior exposure to *Schistosoma* spp. and other infectious agents as well as administration of chemotherapeutic treatment need to be assessed, including baseline immunological and genetic population profiles, as they may impact responses and efficacy to vaccines [16]. Poly-parasitic, in utero, past, ongoing, and recurrent infections of uncrossed and possibly also crossed *Schistosoma* spp. alter immunoregulations suggesting that each infection influences subsequent susceptibility, intensity and immune responses to others.

Of interest related to the ShxSm hybrid type could be the potential of the Smp80-GLA-SE/SchistoShield^®^ vaccine candidate to confer robust, balanced Th1/Th2 cross-species prophylactic and therapeutic protection against Sh in addition to Sj as demonstrated in rodents and baboons [128,129]. This observation could be due to 95% and 84% amino acid homology among calpain of Sh and Sj with Sm, respectively. Likewise, an ongoing phase 1 dose-escalation, safety and immunogenicity Sm-TSP-2/Sm-TSP-2Al^®^ trial coupled with a subsequent 2b Sm-TSP-2/Sm-TSP-2Al^®^ trial among Sm and Sh exposed healthy male and female Ugandan adults receiving three-dose immunization could reveal promising cross-species protective effects depending on the level of inter- and/or intra-species polymorphism of the TSP-2 EC2 domain [130]. Of note is that individuals with putative resistance to Sm despite years of exposure have significantly higher anti-Sm-TSP-2 IgG1 and IgG3 antibody levels than chronically infected [31,32]. An ortholog of Sm TSP-2 among Sj shares >90% sequence identity over the first 116 amino acids indicating TSP-2′s potential of cross-species protection [29]. However, high sequence divergence among the TSP-2 EC2 domain of both species, and an even higher divergence among the EC2 loop of Sj is reported [31]. Cupit et al. stated that this polymorphism could be related to the species’ host spectra as the TSP-2 EC2 domain interacts and complexes with other proteins, including major histocompatibility complex molecules, to evade host immune responses [31].

## 4. Conclusions

Schistosomal hybrids are predominantly reported from West Africa with scattering to Europe, Central and Eastern Africa. However, more sensitive, high-throughput field-suitable diagnostic tools are needed to avoid underdetection of hybrid schistosomes impacting prevention and control measures.

Of note is the dominance of Sh interbreeding with human and veterinary species leading due to hybrid vigor to species extinction and homogenization as seen for Sg in Cameroon and Sh in Niger, respectively, and to species recurrence as observed for Sh and Sb in Corsica with a likely West African origin based on COX1 haplotypes. Heterosis abetted by behavioral, reproductive and genetic advantages seems to predestine ShxSb interbreeds with dominant Sh-ITS/Sb-COX1 pattern to spread across West Africa and beyond. Altered ova excretion solely through the urinary route as detected for ShxSg and ShxSm recombinants among Cameroonian and Beninese, and Côte d’Ivoirian children, respectively, challenges clinical and laboratory diagnostics. Instant acquisition of genetic material among species within the same monophyletic clade of low phylogenetic distance seems likely, thus ShxSm interclade hybrids as seen among Senegalese and Côte d’Ivoirian children are unexpected. Additionally, pure Sb and SbxSc crosses capable of infecting humans through zoonotic transmission found in Corsica and Côte d’Ivoire, and Niger, respectively, are worrisome and demand prevention and control measures targeting human and veterinary sectors in line with the One-Health concept.

Ultimately, a vaccine as the current candidate in advanced to pre-clinical and clinical phases with cross-species protective potential is needed to complement preventive measures, and protect long-term against transmission, infection, and disease recurrence and sequelae. Despite the strong evidence of naturally occurring schistosomal hybrids of large functional and structural diversity, its potential impact on current vaccine candidates is yet unclear and requires further research. Vaccine targets exposed at the schistosome tegumental surface should not evolve rapidly leading to sequence heterogenicity in natural parasite populations, and rendering its use as a vaccine candidate ineffective, but rather highly conserved. Additionally, natural history studies seem of particular importance in areas with known endemicity for hybrid schistosomes that are targeted for clinical trials.

## Figures and Tables

**Figure 2 microorganisms-09-01465-f002:**
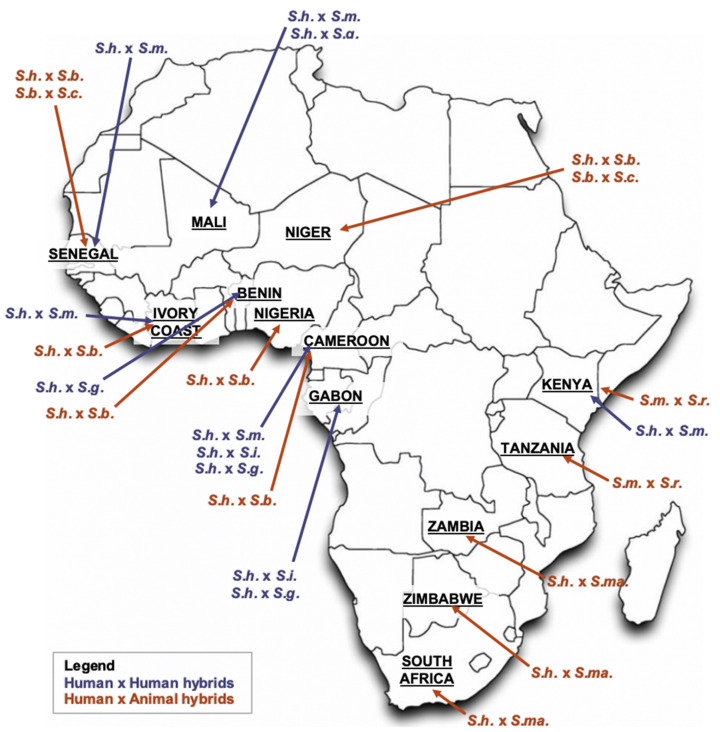
Distribution of Schistosoma hybrids across Africa. Notes: S.h. = *S. haematobium*; S.m. = *S. mansoni*; S.g. = *S. guineensis*; S.i. = *S. intercalatum*; S.b. = *S. bovis*; S.c. = *S. curassoni*; S.r. = *S. rodhaini*; S.ma. = *S. mattheei*; adapted primarily from Leger et al. [5].

**Figure 3 microorganisms-09-01465-f003:**
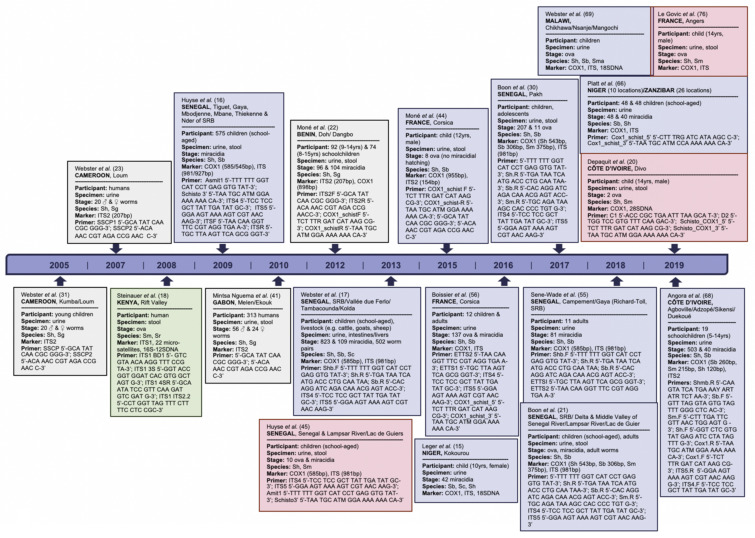
Articles on hybrid *Schistosoma* included with details on study participants, specimens and schistosomal stages investigated, and confirmatory markers applied by chronological order of publication date. Articles highlighted in light grey, green, blue, and red concern ShxSg, SmxSr, ShxSb/Sma/Sc, and ShxSm hybrids, respectively. We searched databases of PubMed and Embase for suitable publications on natural human hybridale schistosomes by applying the following terms: “schistosomiasis”, “Schistosoma”, “snail fever”, “bilharzia”, “introgression”, “hybrid” and “hybridization/hybridisation”. The last searches were performed on 20 April 2021. Publications included after removing duplicates, screening titles and abstracts, reading full-texts, and complementing through reference searches were not restricted by time period, and African setting since extended to Europe; they were limited to full-text availability in English, and hybrid confirmation by mitochondrial cytochrome c oxidase (COX) and/or nuclear ribosomal internal transcribed spacer (ITS). Animal studies, reviews, mathematical models, reports on acute infections from non-endemic areas and travelers, and hybrid confirmation solely by morphological, biological and biochemical characteristics, and experimental species crossing, including snail compatibility studies, were excluded. Notes: *S.* = *Schistosoma*, yrs = years, bp = base pair, NA = not available, ♂ = male, ♀ = female, ITS1 = first nuclear ribosomal internal transcriber spacer, ITS2 = second nuclear ribosomal internal transcriber spacer, ITS = first and second nuclear ribosomal internal transcriber spacer, COX1 = partial mitochondrial cytochrome c oxidase subunit I, 18SDNA = partial nuclear ribosomal 18SDNA, 28SDNA = partial nuclear ribosomal 28SDNA, 12S-16SDNA = partial nuclear mitochondrial 12S-16SDNA, Sh = *S. haematobium*, Sm = *S. mansoni*, Sg = *S. guineensis*, Sr = *S. rodhaini*, Sb = *S. bovis*, Sma = *S. mattheei*, Sc = *S. curassoni*, SRB = Senegal River Basin.

## Data Availability

Not applicable.

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
