# Peer review of "Natural Intra- and Interclade Human Hybrid Schistosomes in Africa with Considerations on Prevention through Vaccination"

_microorganisms, 2021, doi:10.3390/microorganisms9071465_

Round 1
Reviewer 1 Report
1) This review will be absolutely fruitful not only for individual human health issues, but also for becoming great model of the One-Health concept.
2) However, it seems to difficult to understand the relationship between the hybridization of the schistosomes and its control measures, especially vaccination. Hence, the present referee thinks it enough that the systematic summarization of the papers published .
3) By the way, the authors have collected the data derived ONLY from the PubMed and Embase. It is recommended that the other database, e.g., BIOSIS including Zoological Record, is relatively reliable, because the schistosome worms are interesting materials for non-med. or non-vets, viz., zoologists as well.
Reviewer 2 Report
The pesented review manuscrit contains review of current literature on the occurence of schistosomiasis, treatment and prevention. It schows the current state of knowledge on the formation and spread of hybrids Schistosoma spp. and the threats they may pose to human health. This fact is important from the epidemiological point of view. Moreover, the formation of such hybrids may have a potential impact on the effectiveness of currently developed vaccines against these flukes. In my opinion this manuscript is developed reliably and contains correctly drawn conclusions. It may be published in Microorganism Journal in the present form, but before publications authors should make minor adjustment, see below:
- Line 292, „spp” should be not written in italics.
- Line 425 there is „ShxSma-18SDNAxSma-COX1”. I suggest put an explanation of the shortcut Sma (S. mattheei) somewhere nearby.
